# Effects of Different Concentrations of Micro-Nano Bubbles on Grain Yield and Nitrogen Absorption and Utilization of Double Cropping Rice in South China

Yinfei Qian [ID], Xianjiao Guan, Caihong Shao, Caifei Qiu, Xianmao Chen, Jin Chen [ID] and Chunrui Peng *

Soil and Fertilizer & Resources and Environmental Institute, Jiangxi Academy of Agricultural Sciences, Nanchang 330200, China
* Correspondence: pcrtfs@163.com; Tel./Fax: +86-791-8709-0358

**Abstract:** Micro-nano bubble (MNB) irrigation can effectively improve the hypoxia stress caused by conventional irrigation and shows great potential in plant development, yield improvement, and saving of water and fertilizer, and has been recognized as a new and high-efficiency technology in crop planting. However, former research on MNB concentration had no clear segmentation, and other MNB concentrations can achieve better or worse effects. This remains to be further explored in order to explore the optimal concentration of MNBs for the yield and nitrogen absorption and utilization of the double cropping rice. With early rice Ganxin203 and late rice WufengyouT025 as the experimental cultivars, the effects of MNBs on growth, yield, and nitrogen absorption and utilization of the potted double cropping rice were analyzed by setting three concentrations of MNBs (LM, low concentration; MM, middle concentration; HM, high concentration), compared with the ordinary running water (CK). Compared with CK, grain yield of the early rice under the MNB treatment increased by 4.84~10.95% and the late rice increased by 6.10~14.31%. It was found that the higher the concentration of the MNBs, the higher the yield of the rice. This is due to that the MNBs improved the tiller-bearing rate, increased the SPAD and Pn values of the flag leaves in the whole growth period, slowed down the drop of the leaf SPAD and Pn from heading stage to maturity, increased the number of the adventitious roots, improved the α-NA oxidation of the root, and simultaneously promoted the nitrogen accumulation, absorption, and utilization. The HM treatment obtained the best benefits, and the effect of the MNBs on the late rice was better than the early rice.

**Keywords:** micro-nano bubbles; double cropping rice; root characteristics; grain yield; nitrogen absorption and utilization

## 1. Introduction

Rice (*Oryza sativa* L.) is the world's most important food crop and a primary source of food for more than half the world's population [1]. How to achieve high yields has become a major goal of researchers. For a long time, high rice yield was mainly achieved through large amount of fertilizers and water [2]. Excessive fertilizer and water use not only increases the cost of rice production, but also easily leads to some serious environmental problems [3], such as water eutrophication, soil acidification, greenhouse gas emissions, and so on. In order to solve these problems, in addition to innovating the traditional cultivation technology, the application of high-yield cultivation technology without any pollution has attracted more and more attention [4].

According to the study of Li et al. [5], the soil oxygen content for rice roots to maintain normal physiological activities is 3–5%, and root elongation stops when the root surface oxygen concentration is lower than 0.001 mol·m$^{-3}$. Rice requires irrigation, and conventional irrigation often leads to rhizosphere oxygen being discharged by soil water as well as root zone hypoxia, which causes rice to grow slowly and even reduce grain yield. To coordinate the contradiction between water and oxygen, scholars put forward the method

of oxygenated irrigation, that is, adding oxygen to irrigation water, so as to avoid problems such as growth retardation caused by root zone hypoxia due to irrigation to improve crop growth [6]. Micro-nano bubble (MNB) irrigation is one type of oxygenated irrigation and has shown great potential in saving water and fertilizer and improving yield in recent years [7,8]. The diameter of MNBs ranges from 50 μm to 200 nm. MNBs have quite a large specific surface area and strong gas dissolving capacity that can transfer the gas into the water quickly [9]; usage of MNBs has become the most highly effective aeration method [10]. MNBs may rapidly enhance the dissolved oxygen in irrigation water, and they also have high Zeta potential, which can hinder their gathering and allow them to persist for a long time, even for several months, in water [11]. Simultaneously, the volume of the MNBs is small and the surface area is large, which make the MNBs easily adhere to the crops' roots and strengthen the adsorption ability of the nutrients [8].

At present, the application of MNBs in crop production is mainly in seed soaking and irrigation [12,13]. Ahmed et al. [12] reported that the seeds in water-containing NBs exhibited 6–25% higher germination rates. Jiang et al. [14] found that the MNBs improved the germination rate, germinating energy, and vigor index of the tested vegetables. Many research studies have focused on the effect of MNB irrigation on hydroponic greenhouse vegetables (leaf lettuce, cabbage, and edible rape), sugarcane, tomato, and cucumber [15,16], and found that the MNBs can enhance the yield of them easily. There are only a few reports on the effect of MNBs on paddy rice. Cai et al. [17,18] reported that MNB irrigation can improve the yield of rice and the water use efficiency as well as reduce the runoff of nitrogen and phosphorus. Sang et al. [19] found that MNBs are helpful in reducing nitrogen losses, enhancing the output of early rice, and improving the nitrogen use efficiency. Wang et al. [20] reported that MNBs promote nutrient utilization and plant growth in rice by upregulating nutrient uptake genes and stimulating growth hormone production. However, these experiments of MNB concentrations had no clear segmentation, and other MNB concentrations can achieve better or worse effects on the growth of paddy rice, especially in double cropping rice. Double cropping rice is a typical farming system in Southern China, accounting for 1/3 of the rice planting area in China in 2021 [21], and China accounts for about 20% of the world's rice acreage. The double cropping rice system typically consists of early rice (from March to July) and late rice (from June to October), and the optimal MNB concentration for double cropping rice is not yet known. Will high concentrations of MNBs lead to negative effects on rice growth? This remains to be further explored.

In this study, we adjusted different air admission speeds to form different MNB concentrations and conducted a two-year potted experiment to investigate the characteristics of growing and nitrogen absorption and utilization of the double cropping rice under different MNB concentrations. Our main objectives were (1) to investigate the effect of the different MNB concentrations on the agronomical, physiological, and nutritional traits of the double cropping rice and (2) to investigate which is the optimal MNB concentration for the double cropping rice.

## 2. Materials and Methods

### 2.1. Site Descriptions

The experiment was conducted at the greenhouse of the Jiangxi Academy of Agricultural Sciences, located in Nanchang County, Jiangxi province, China (N 28°33′, E 115°56′, 25 m asl), in the double cropping rice-growing seasons in 2018–2019 from March to October. This site has a typical subtropical climate with an average annual temperature is 17.8 °C. The annual precipitation and evaporation of this region are 1662.5 mm and 1610.4 mm, respectively. The annual frost-free period reaches 280 days. The paddy soil of this experiment is yellow clayey soil that developed from the soil parent material of quaternary red clay. The soil properties before the experiment were as follows: soil organic carbon (SOC) 18.4 g kg$^{-1}$, total nitrogen (TN) 1.56 g kg$^{-1}$, total phosphorus (TP) 0.48 g kg$^{-1}$, total potassium (TK) 18.2 g kg$^{-1}$, alkali solution nitrogen (AN) 134 mg kg$^{-1}$, available

phosphorus (AP) 31.2 mg kg$^{-1}$, available potassium (AK) 72.4 mg kg$^{-1}$, cation exchange capacity (CEC) 7.21 cmol kg$^{-1}$, pH 5.42. The early rice cultivar Ganxin203 and the late rice cultivar WufengyouT025 were used in this experiment. They are all super hybrid indica rice widely cultivated in South China. The micro-nano bubble generator used in this study was the screw micro-nano bubble pump (XPK-0.75, Beijing University of Chemical Technology, Peking, China), the working parameters are: the voltage was 380 V, the electric power was 0.75 KW, the diameters of the bubbles were from 700 nm to 12 μm, the speed of the raising bubbles was from 10–15 mm s$^{-1}$, the gas content was 82–90%.

## 2.2. Experimental Design and Crop Management

The experiment consisted of four treatments: (i) ordinary running water, marked as CK; (ii) low-concentration micro-nano meter bubbles, the air admission speed was 300 mL min$^{-1}$, marked as LM; (iii) middle-concentration micro-nano meter bubbles, air admission speed was 600 mL min$^{-1}$, marked as MM; (iv) high-concentration micro-nano meter bubbles, air admission speed is 900 mL min$^{-1}$, marked as HM. There were 30 replicates per treatment. The different concentrations of micro-nano meter bubbles produced by the screw micro-nano bubble pump were achieved through adjusting the air flowmeter (the air admission range was 100–1000 mL min$^{-1}$, 20 °C, 101.325 kPa) to control the speed of air admission and reach the stable setting oxygen concentration. The dissolved oxygen was measured by the Dissolved Oxygen Sensor (YSI 550A, Xylem Inc., STATE OF OHIO, Washington, DC, USA). The water management was added every 5 days from rice transplantation to maturity. The water volume was precisely measured by a graduated container to ensure the same quantity. The changes in dissolved oxygen (DO) concentration in soil solution under different treatments after rice transplanting are shown in Figure 1. The average DO concentration of CK, LM, MM, and HM in the soil solution was 5.24, 5.67, 6.3, and 6.86 mg L$^{-1}$, respectively, in the rice-growing period.

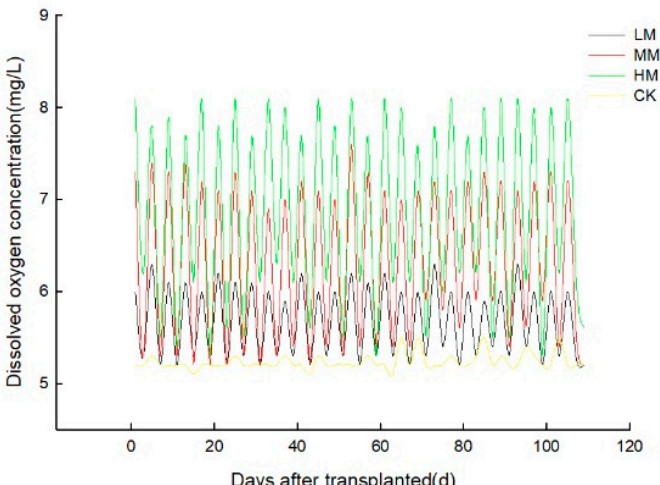

**Figure 1.** Dissolved oxygen (DO) concentration changes in soil solution under different treatment after rice transplanting. LM, MM, HM, and CK represent low-concentration MNBs, middle-concentration MNBs, high-concentration MNBs, and the ordinary running water, respectively. The same as below.

Seedlings were dry-nursed in plastic plates and transplanted in circular plastic pots: the diameter of the pot was 25 cm, the height was 30 cm. In 2018 the early rice was sown on 20 March, transplanted on 20 April, and harvested on 15 July, and the late rice was sown on 20 June, transplanted on 25 July, and harvested on 25 October. In 2019 the early rice was sown on 12 April, transplanted on 30 April, and harvested on 18 July, and the late rice was sown on 20 June, transplanted on 25 July, and harvested on 30 October. Similar healthy seedlings were chosen before transplantation. Each plastic pot contained 18 kg dry rice-field soil plus urea–potassium chloride–calcium magnesium phosphate fertilizer (3.0 g–1.32 g–5.7 g for early rice and 4.5 g–1.98 g–8.55 g for late rice, respectively), and

fertilizers were all applied as basal fertilizer 1 day before rice seedling transplantation. There were 3 seedlings per hole and 3 holes per pot for both the early and late rice. Weeds, insects, and disease were controlled as required, to avoid yield loss. Rain was avoided in the whole rice-growing period.

### 2.3. Sampling and Measurements

### 2.3.1. Observation of Tiller Characteristics

The number of tillers were counted at weekly intervals from transplanting to harvest. Tiller-bearing rate was defined as the ratio of panicle number developed from tillers to the maximum tiller number.

### 2.3.2. Measurement of Leaf SPAD Value and Net Photosynthetic Rate (Pn) Value

The flag leaf SPAD value and net photosynthetic rate were measured from 5 pots each treatment at the jointing stage, heading stage, and maturity stage. The leaf SPAD value was measured by Chlorophyll Meter (SPAD-502, Minolta, Tokyo, Japan) and the mean data of 5 pots were recorded. Net photosynthetic rate was measured at 9:00–11:00 a.m. with the portable Li-6400XT (Li-COR, Lincoln, NB, USA). All the measurements were taken at the saturation irradiance with an incident photosynthetic photo flux density (PPFD) of 1200 $\mu$mol m$^{-2}\cdot$s$^{-1}$ and an airflow rate at 500 $\mu$mol s$^{-1}$.

### 2.3.3. Observation of Root Characteristics

At the full heading stage, 3 representative pots from each treatment were chosen to measure the root characteristics. The roots were carefully rinsed until achieving a clean appearance with a hydropneumatic elutriation device and detached from their nodal basis. The adventitious root number was counted, and the longest root length and the root diameter were measured by vernier caliper. The root system volumes were measured by metering cylinder using the drainage method. The root oxidation activity was determined by measuring oxidation of alpha-naphthylamine ($\alpha$-NA) according to the method of Qiao [22].

### 2.3.4. Biomass Observation

At the maturity stage, 5 pots of rice plants were sampled from each treatment to observe the biomass accumulation. The rice plants were separated into roots, leaves, stems, and panicles. Then, all parts were desiccated at 105 °C and oven-dried at 80 °C to constant weight for determining the biomass.

### 2.3.5. Grain Yield and Its Components

Rice grains were manually harvested at the maturity stage from 5 pots sampled randomly from each treatment. Grain yield and filled grain weight were measured and adjusted to 14% moisture. The number of panicles per pot, the number of grains per panicle, and the percentage of filled grains were counted.

### 2.3.6. Nitrogen Observation

The biomass at maturity for each plant was powdered. The nitrogen concentration was determined by the methods described by Motsara and Roy [23]. Nitrogen accumulation was calculated from the corresponding nitrogen concentration and the biomass.

PFPN, partial factor productivity of N; PFPN (kg kg$^{-1}$) = grain yield/N application rate.

### 2.4. Statistical Analysis

Since year was not a significant factor and the rules of the changes were similar in this experiment during the year 2018 and 2019, we mainly used the data of the year 2018. All statistical analyses were performed using SPSS 19.0 (SPSS Inc., Chicago, IL, USA). A one-way analysis of variance (ANOVA) was used to determine if there were significant differences or not among treatments. Differences among means were separated by using

the least significant difference (LSD) test. Graphs were drawn with Sigma Plot 12.5 (Systat Software Inc., San Jose, CA, USA).

## 3. Results

### 3.1. Grain Yield and Its Components

As shown in Table 1, the grain yield changes in the double cropping rice showed the same tendency both in the year 2018 and 2019. The yields of different micro-nano bubble concentration treatments were significantly higher than that of the ordinary running water (CK) treatment. The LM, MM, and HM treatment of the early rice Ganxin203 were higher than CK by 5.43%, 7.77%, and 10.95% in 2018 and 4.84%, 9.19%, and 10.87% in 2019, respectively. The LM, MM, and HM treatments of the late rice WufengyouT025 were higher than CK by 6.10%, 12.81%, and 14.31% in 2018 and 7.05%, 11.32%, and 13.37% in 2019, respectively. The grain yield of the LM, MM, and HM treatments increased along with the increasing concentrations of the MNBs. The application of MNBs in the late rice season was better than that in the early rice season. The number of panicles, grains per panicle, and filled grains were increased along with the ascension of the MNB concentration. The grains per panicle was the most affected by the concentration of MNBs, followed by the panicle and filled grains. The filled grain weight was the least affected by the concentration of MNBs, and the difference was not remarkable.

**Table 1.** Grain yield and yield components of the double cropping rice under different treatments.

| Year | Type | Treatment | Panicles (no pot$^{-1}$) | Grains per Panicle | Filled Grains (%) | Filled Grain Weight (mg) | Grain Yield (g pot$^{-1}$) |
|---|---|---|---|---|---|---|---|
| 2018 | Early rice | CK | 21.3 ± 0.4 a | 105.5 ± 2.8 b | 86.32 ± 0.42 c | 27.72 ± 0.26 a | 50.68 ± 2.53 b |
| | | LM | 21.5 ± 0.5 a | 106.6 ± 3.4 b | 87.04 ± 0.26 b | 27.74 ± 0.15 a | 53.43 ± 3.46 ab |
| | | MM | 21.5 ± 0.8 a | 109.4 ± 2.2 ab | 87.25 ± 0.54 ab | 27.75 ± 0.18 a | 54.62 ± 2.6 ab |
| | | HM | 21.7 ± 1.2 a | 112.2 ± 1.4 a | 87.33 ± 0.42 a | 27.74 ± 0.12 a | 56.23 ± 1.58 a |
| | Late rice | CK | 21.2 ± 0.4 c | 164.4 ± 0.9 d | 82.55 ± 0.4 b | 23.1 ± 0.34 a | 64.23 ± 2.62 b |
| | | LM | 21.6 ± 0.6 b | 169.2 ± 2.9 c | 83.26 ± 0.78 a | 23.14 ± 0.25 a | 68.15 ± 2.63 ab |
| | | MM | 21.8 ± 0.5 ab | 176.1 ± 3.4 b | 83.43 ± 0.83 a | 23.14 ± 0.14 a | 72.46 ± 3.92 a |
| | | HM | 22 ± 0.8 a | 180.3 ± 2.1 a | 83.54 ± 0.65 a | 23.14 ± 0.18 a | 73.42 ± 2.24 a |
| 2019 | Early rice | CK | 25.5 ± 0.4 b | 110.4 ± 2.8 d | 87.26 ± 0.33 c | 27.52 ± 0.32 a | 65.32 ± 1.53 b |
| | | LM | 25.6 ± 0.2 b | 114.5 ± 3.3 c | 88.13 ± 0.62 b | 27.54 ± 0.24 a | 68.48 ± 2.87 ab |
| | | MM | 25.8 ± 0.4 a | 118.2 ± 4.2 b | 88.56 ± 0.34 a | 27.51 ± 0.28 a | 71.32 ± 3.88 a |
| | | HM | 25.9 ± 0.3 a | 122.1 ± 3.2 a | 88.84 ± 0.54 a | 27.57 ± 0.06 a | 72.42 ± 2.62 a |
| | Late rice | CK | 24.3 ± 0.3 b | 176.5 ± 3.2 d | 82.43 ± 0.25 d | 23.52 ± 0.13 a | 80.45 ± 2.2 b |
| | | LM | 24.6 ± 0.8 ab | 185.6 ± 4.3 c | 83.17 ± 0.32 c | 23.54 ± 0.24 a | 86.12 ± 2.32 a |
| | | MM | 24.8 ± 0.4 a | 189.2 ± 5.7 b | 83.37 ± 0.36 b | 23.58 ± 0.33 a | 89.56 ± 3.36 a |
| | | HM | 25 ± 0.6 a | 193.6 ± 4.1 a | 83.63 ± 0.43 a | 23.6 ± 0.37 a | 91.21 ± 2.43 a |

Note: Values within a column followed by a different letter are significantly different at 5% (lowercase letter) probability levels. LM, MM, HM, and CK represent low-concentration MNBs, middle-concentration MNBs, high-concentration MNBs, and the ordinary running water, respectively. The same as below.

### 3.2. Tiller Characteristics

Figure 2A showed that the max tillers of the double cropping rice increased with the concentration of MNBs, but the difference was not significant, while the tiller-bearing rate (Figure 2B) increased with the MNB concentration significantly. The tiller-bearing rate of the LM, MM, and HM treatment of the early rice was 1.43, 2.29, and 6.22 percentage points higher than CK, respectively, while the late rice was 4.19, 8.09, and 10.55. The difference reached the significant level. It can be seen that MNBs mainly improve the quality of tillers rather than the quantity of tillers. The effect of MNBs on the tiller characteristics of late rice was better than that of early rice.

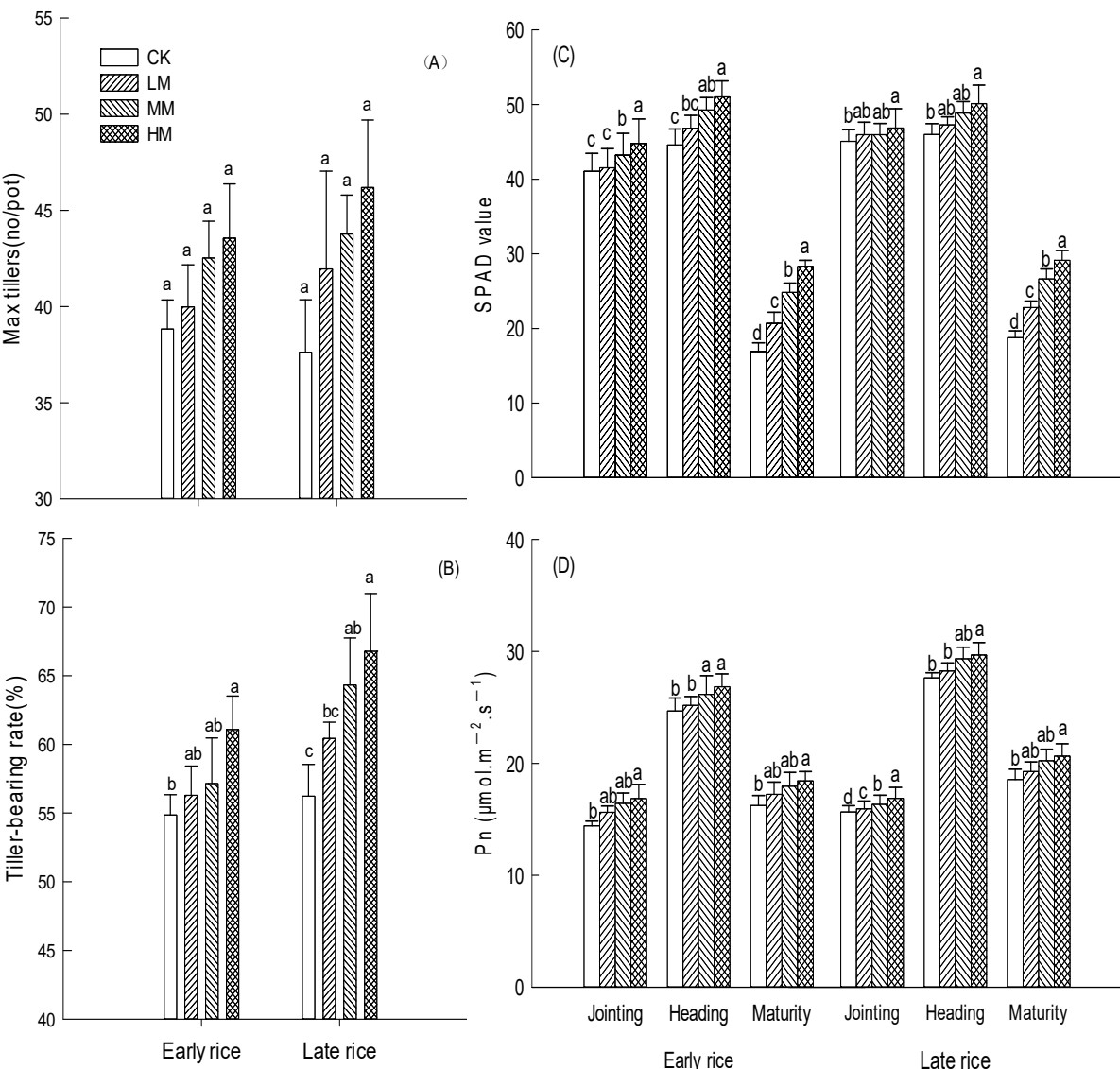

**Figure 2.** Max tillers (**A**), tiller-bearing rate (**B**), SPAD (**C**), and Pn (**D**) of the double cropping rice under different treatments (2018). LM, MM, and HM denote low-, middle-, and high-concentration MNBs, respectively. CK denotes the ordinary running water. The vertical bars represent $\pm$ SE of the mean. The SE was calculated across three replications. Different letters indicate statistical significance at $p < 0.05$.

### 3.3. Photosynthesis Characteristics

The SPAD and Pn values are two important parameters, as these leaf photosynthesis characteristics measure the chlorophyll content and net photosynthetic rate, respectively. The SPAD and Pn value of the flag leaf of the double cropping rice during different growing stages are shown in Figure 2. The SPAD (Figure 2C) and Pn (Figure 2D) values of the flag leaf rose first and reached the maximum value at the heading stage and then decreased. The SPAD and Pn values of rice increased along with the MNB concentration increasing. From heading to maturity, the decreased rates of the SPAD values of the treatment of LM, MM, and HM of the rice were 10.17%, 20.10%, and 28.25% slower than CK (early rice) and 12.55%, 23.12%, and 29.25% slower than CK (late rice), respectively. The decreased rates of the Pn values of the treatment of LM, MM, and HM of the rice were 7.42%, 8.09%, and 8.12% slower than CK (early rice) and 3.27%, 5.52%, and 7.42% slower than CK (late rice), respectively. This indicates that increasing the concentration of MNBs can slow down the leaf senescence effectively.

### 3.4. Root Characteristics

There were significant differences in root characteristics between early rice and late rice at the heading stage (Table 2). Late rice had a better root characteristic than early rice. The MNBs promoted the root growth of the double cropping rice. Along with the MNB concentration increase, the double cropping rice's adventitious root number, the longest root length, root dry matter weight, root volume, and $\alpha$-NA oxidation increased, while the root diameter decreased. The $\alpha$-NA oxidation is an important indicator of physiological metabolism in rice. High $\alpha$-NA oxidation is beneficial to maintain the root system function and the growth of rice plants. The $\alpha$-NA oxidation of the LM, MM, and the HM treatment of the double cropping rice increased 4.06%, 14.68%, and 25.74% over CK (early rice) and 10.38%, 16.75%, and 21.6% over CK (late rice). The $\alpha$-NA oxidation of the late rice was higher than that of the early rice under the same MNB concentration.

**Table 2.** Root characteristics of the double cropping rice of different MNB concentrations at the heading stage.

| Type | Treatment | Number of Adventitious Roots | The Longest Root Length (cm) | Root Diameter (mm) | Root Dry Matter (g pot$^{-1}$) | Root Volume (cm$^3$ pot$^{-1}$) | $\alpha$-NA Oxidation ($\mu$g h$^{-1}$ g$^{-1}$) |
|---|---|---|---|---|---|---|---|
| Early rice | CK | 208 ± 13 b | 31.1 ± 4.3 b | 0.91 ± 0.103 a | 5.58 ± 0.83 c | 72.48 ± 3.28 c | 54.43 ± 1.34 c |
| | LM | 224 ± 21 ab | 35.4 ± 2.8 ab | 0.88 ± 0.064 a | 7.42 ± 2.13 bc | 122.33 ± 9.39 b | 56.64 ± 2.12 c |
| | MM | 245 ± 25 ab | 37.2 ± 4.3 ab | 0.86 ± 0.045 a | 10.34 ± 1.56 b | 142.68 ± 14.56 b | 62.42 ± 1.87 b |
| | HM | 268 ± 31 a | 38.6 ± 2.3 a | 0.84 ± 0.031 a | 13.32 ± 2.76 a | 178.2 ± 13.98 a | 68.44 ± 2.32 a |
| Late rice | CK | 245 ± 23 b | 35.2 ± 2.4 b | 1.02 ± 0.13 a | 6.84 ± 0.64 c | 120.45 ± 5.86 c | 60.23 ± 1.21 c |
| | LM | 263 ± 25 ab | 38.5 ± 3.8 ab | 0.99 ± 0.102 a | 8.28 ± 1.21 c | 145.78 ± 12.39 bc | 66.48 ± 0.94 b |
| | MM | 288 ± 32 ab | 40.5 ± 2.8 ab | 0.93 ± 0.098 a | 11.56 ± 0.89 b | 168.54 ± 4.2 ab | 70.32 ± 5.72 ab |
| | HM | 309 ± 24 a | 42.4 ± 3 a | 0.92 ± 0.045 a | 13.88 ± 2.21 a | 188.6 ± 32.15 a | 73.24 ± 3.68 a |

Note: Values within a column followed by a different letter are significantly different at 5% (lowercase letter) probability levels.

### 3.5. Biomass

The effects of different MNB concentrations on the biomass of the double cropping rice are shown in Table 3. The biomass of the whole parts of the rice including leaves, stems, ears, and roots of the double cropping rice as well as the ratio of the root/shoot increased with the concentration of MNBs. The HI (harvest index) changed very little, and the difference is not obvious. The total biomass of LM, MM, and HM increased 4.75%, 9.54%, and 13.92% over CK (early rice) and 7.12%, 14.59%, and 17.95% over CK (late rice), respectively.

**Table 3.** Biomass of double cropping rice of different treatments at maturity (2018).

| Type | Treatment | Biomass (g pot$^{-1}$) | | | | | Root/Shoot | HI |
|---|---|---|---|---|---|---|---|---|
| | | Leaf | Stem | Ear | Root | Total | | |
| Early rice | CK | 16.92 ± 0.94 b | 25.34 ± 1.12 b | 52.34 ± 1.63 c | 6.02 ± 0.24 d | 102.62 ± 3.93 c | 0.062 ± 0.000 d | 0.52 ± 0.02 a |
| | LM | 17.86 ± 1.45 ab | 25.95 ± 0.42 ab | 56.32 ± 2.55 bc | 7.36 ± 0.12 c | 107.59 ± 4.54 bc | 0.074 ± 0.003 c | 0.53 ± 0.03 a |
| | MM | 18.26 ± 1.31 ab | 26.65 ± 0.32 a | 58.42 ± 1.78 ab | 9.08 ± 0.28 b | 112.41 ± 3.49 ab | 0.087 ± 0.003 b | 0.53 ± 0.01 a |
| | HM | 19.48 ± 1.46 ab | 26.27 ± 0.46 ab | 60.42 ± 2.23 a | 10.73 ± 0.32 a | 116.9 ± 4.47 a | 0.10 ± 0.004 a | 0.53 ± 0.02 a |
| Late rice | CK | 23.22 ± 1.12 b | 28.34 ± 0.65 b | 68.44 ± 1.34 c | 4.74 ± 0.21 d | 124.74 ± 3.32 c | 0.041 ± 0.001 c | 0.54 ± 0.00 a |
| | LM | 24.66 ± 1.25 ab | 30.18 ± 0.58 a | 72.34 ± 1.92 b | 6.46 ± 0.54 c | 133.64 ± 4.29 b | 0.041 ± 0.003 c | 0.54 ± 0.01 a |
| | MM | 25.68 ± 1.36 a | 32.34 ± 0.48 a | 76.68 ± 2.45 a | 8.26 ± 0.14 b | 142.96 ± 4.43 a | 0.048 ± 0.004 b | 0.54 ± 0.00 a |
| | HM | 25.54 ± 1.13 a | 32.89 ± 0.5 a | 77.58 ± 1.6 a | 11.15 ± 0.31 a | 147.16 ± 3.54 a | 0.06 ± 0.003 a | 0.54 ± 0.02 a |

Note: Values within a column followed by a different letter are significantly different at 5% (lowercase letter) probability levels.

### 3.6. Nitrogen Absorption and Utilization

The nitrogen concentration of leaves, stems, grains, and roots of double cropping rice increased with MNB concentration, but the difference was small (Figure 3A).

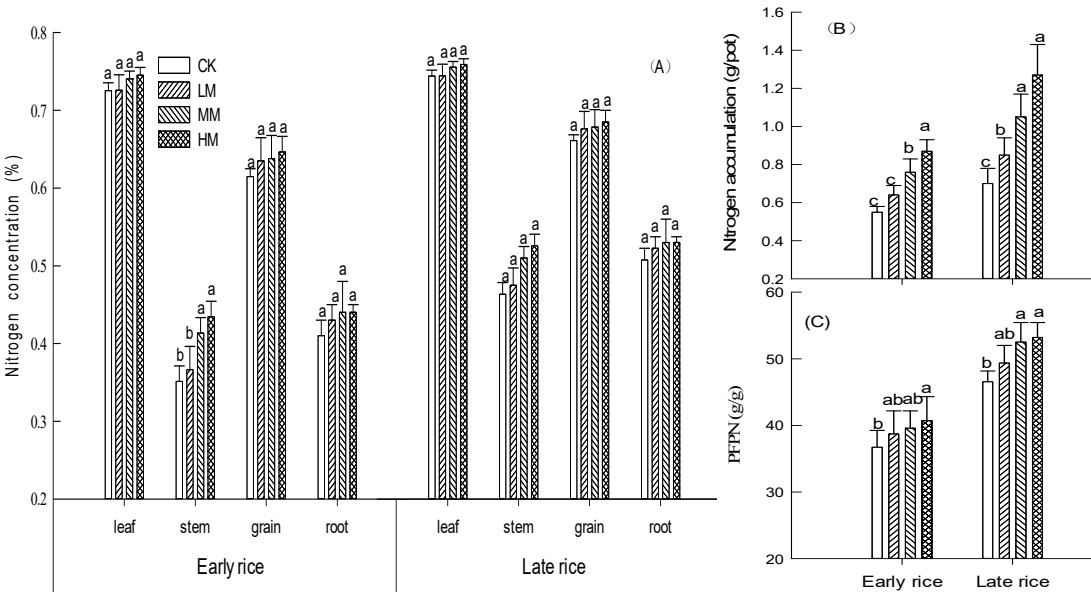

**Figure 3.** Nitrogen concentration (**A**), nitrogen accumulation (**B**), and PFPN (**C**) of the double cropping rice under different treatments (2018). LM, MM, and HM denote low-, middle-, and high-concentration MNBs, respectively. CK denotes the ordinary running water. The vertical bars represent ± SE of the mean. The SE was calculated across three replications. Different letters indicate statistical significance at $p < 0.05$.

### 3.7. Correlation Analysis of Rice Characteristics with Grain Yield

Correlation analysis showed a significant positive correlation among total biomass, N accumulation amount, PFPN, and grain yield (r = 0.973 * − 0.987 *) of the early rice. There was significant or extremely significant positive correlation among α-NA oxidation, total biomass, tiller-bearing rate, N accumulation amount, PFPN, and yield of the late rice (r = 0.952 ** − 0.997 **) (Table 4).

**Table 4.** Correlation coefficients of grain yield with main characteristics of the double cropping rice of different treatments (2018).

| Type | Characteristics | Grain Yield | α-NA Oxidation | Total Biomass | Tiller-Bearing Rate | N Accumulation | PFPN |
|------|-----------------|-------------|----------------|---------------|---------------------|----------------|------|
| | Grain yield | 1 | | | | | |
| | α-NA oxidation | 0.934 | 1 | | | | |
| Early rice | Total biomass | 0.987 * | 0.978 * | 1 | | | |
| | Tiller-bearing rate | 0.912 | 0.965 * | 0.939 | 1 | | |
| | N accumulation | 0.973 * | 0.990 ** | 0.997 ** | 0.949 | 1 | |
| | PFPN | 0.977 * | 0.978 * | 0.997 ** | 0.92 | 0.996 ** | 1 |
| | Grain yield | 1 | | | | | |
| | α-NA oxidation | 0.989 * | 1 | | | | |
| Late rice | Total biomass | 0.997 ** | 0.994 ** | 1 | | | |
| | Tiller-bearing rate | 0.993 ** | 0.996 ** | 0.999 ** | 1 | | |
| | N accumulation | 0.956 * | 0.967 * | 0.975 * | 0.982 * | 1 | |
| | PFPN | 0.952 * | 0.973 * | 0.948 | 0.949 | 0.886 | 1 |

Note: * Statistical significance at $p < 0.05$. ** Statistical significance at $p < 0.01$.

## 4. Discussion

### 4.1. How MNBs Improve the Grain Yield of the Double Cropping Rice

In this study, MNBs obviously enhanced the grain yield of the double cropping rice. The early rice Ganxin203 increased by 4.84~10.95% and the late rice WufengyouT025 increased by 6.10~14.31%. There were three main reasons for this result.

First, the MNBs enhanced the rhizosphere dissolved oxygen content of the double cropping rice. The previous research [24,25] generally believed that the MNBs can enhance

the dissolution efficiency of oxygen in water, compared with the traditional aeration effect. The MNBs have the advantages of small bubble volumes, large specific surface area, slow rising speed, and long residence time in water. They are "oxygen reservoirs" and can effectively alleviate the hypoxia problem of the rhizosphere soil during the progress of the irrigation. This research also proved this, in which the micro-nano bubble generator produced MNBs which significantly improved the dissolved oxygen content in the rice rhizosphere and maintained it for a long time. Simultaneously, the faster the air admission speed of the micro-nano bubble generator was, the higher the dissolved oxygen concentration and the better the oxygenation effect. Oxygen is one of the necessary conditions for higher plants to carry on normal physiological metabolism and growth. It is also the terminal electron receptor in the respiratory chain, driving the synthesis of ATP and NADPH and providing energy for cell growth during oxidative phosphorylation [26]. Although rice is a helophyte, in order to adapt to the flooded environment, the cortical cells in its mature area will undergo programmed death and dissolve and form the aerenchyma, then transport oxygen from the ground to the root to satisfy the root respiration [27]; however, the soil oxygen content is still the essential factor restricting the root growth of rice [28]. When irrigating rice fields, although the water demand of rice is satisfied, the soil air porosity of rice roots is reduced, resulting in the lack of soil oxygen in rice roots. The diffusion rate of oxygen in the outside air is very slow, only about 1/10,000, and the solubility in the water is only 0.5% to 0.7%. Therefore, oxygen in the air has difficultly entering the soil through the water layer, which will cause a lack of oxygen in the rice field and disadvantageously influence the growth of rice [5]. Therefore, it is important to use MNBs to improve the oxygen content of the rice root rhizosphere, reduce the bad effects of hypoxia, improve the growth of rice, and enhance the output of rice [29].

Second, the MNBs probably promoted the biological activity of the double cropping rice. Exogenous reactive oxygen species (ROS) produced by MNBs can promote plant metabolism and growth. Simultaneously, the MNBs produce hydroxyl radicals (.OH), which can enhance the expression of genes associated with catalase and NADPH oxidase [30]. The increase in catalase can promote the decomposition of hydrogen peroxide into water and oxygen, thus reducing the toxicity of hydrogen peroxide to rice. In this study, compared with CK, three MNB concentration treatments enhanced the tiller number and made more tillers, improving the tiller-bearing rate. The MNBs also improved the SPAD and Pn values of the flag leaves. The MNBs increased the adventitious root number, the longest root length, root dry matter weight, root volume, $\alpha$-NA oxidation, and the biomass of all parts of the rice. At the same time, the MNBs also promoted the accumulation, absorption, and utilization of nitrogen, and increased the grain yield. Meanwhile, the hydroxyl radicals produced by MNBs could also delay senescence. In this study, MNBs significantly slowed down the decline of SPAD and Pn values of the flag leaves and the senescence speed of the double cropping rice.

Third, the MNBs may promote the abundance, activity, and function [31,32] of the potential bacteria related to nitrification and nitrogen fixation, enhance the dynamic mutual feedback between soil fertility and microorganisms, and enhance the output of the rice altogether.

For the effect of MNBs on the rice yield components, different research has had different results. Zhu et al. [33] observed that the application of MNBs can improve the rice yield by increasing the number of panicles, grain per panicle, and grain weight. Cai et al. [18] reported that the application of MNBs could improve the number of panicles and grains per panicle of early rice, but reduce the filled grains. This study reported that the application of MNBs significantly increased the grain number per panicle of the double cropping rice, followed by panicle number and filled grains, with little effect on filled grain weight.

This may be related to the fact that the rice varieties used in this experiment were those with large storage capacity and grains that are susceptible to environmental influence. This study also found that the growth, nitrogen absorption, and yield of late rice under the

same concentration of MNBs were significantly better than those of early rice. Whether this is due to the different genotypes of early and late rice or the different external growing environment of early and late rice remains to be discovered in further research.

*4.2. Optimal MNB Concentration for the Double Cropping Rice*

Different plants need different dissolved oxygen concentrations for growth and development [34–36], and so does paddy rice [37,38]. The soil oxygen content for maintaining normal physiological activity is 3% to 5%, and the roots stop elongating when the root surface oxygen concentration is lower than 0.001 mol m$^{-3}$ [5]. Soil oxygen content that is too low may not satisfy the metabolic effect of rice roots, causing slow growth and weak water absorption; soil oxygen content that is too high may make the air fill the soil pores so that water cannot enter, resulting in the root system not being able to absorb water, therefore reducing its water utilization and hindering the growth of roots and aboveground parts. Because MNBs are the carriers of oxygen, it can be inferred that there must exist an optimal MNB concentration to obtain the optimal dissolved oxygen concentration. Meanwhile, according to Park's research [39], vegetables grown in a solution containing MNBs grow faster than vegetables grown in ordinary solution under the same dissolved oxygen content. This indicates that in addition to being oxygen carriers, MNBs also have unique characteristics such as large air–liquid specific surface area, high adsorption efficiency, and long residence time in water, which can change the growth and development processes of plants. Therefore, the optimal MNB concentration and the optimal dissolved oxygen concentration are not same. Liu et al. [40] reported that the reactive oxygen species (ROS) produced by MNBs are positively correlated with plant growth, plant quality, and seed germination. Low-concentration MNB stress will cause the root to transmit a hypoxia signal to the canopy, affecting the transportation and storage of growth matter such as water and auxins, leading to a reduction in plant production. However, when the concentration exceeds the toxicity threshold, the plant itself is subjected to oxidative stress, and the metabolic balance in the plant body is broken. The accumulated reactive oxygen species will cause bad effects on all aspects of the plant, and severe oxidative stress will lead to plant death. Therefore, it is necessary to clarify the optimal MNB concentration for each plant. Zhou et al. [16] reported that 15 mg L$^{-1}$ and fruiting period as the optimal MNB concentration and application period of MNBs to tomato and cucumber. Du et al. [35] used air aeration and had the best effect on the growth and nitrogen absorption and utilization of summer corn at the dissolved oxygen level of 10 mg L$^{-1}$ for 10 min at 0.1 MPa. At the same time, air aeration is cheaper than pure oxygen aeration and easy to operate. Zhou et al. [15] reported the production of aquatic vegetables was first increased and then decreased with the increasing concentration of MNBs, and 10–15 mg L$^{-1}$ MNBs bubbles was the optimal. Zhang et al. [41] reported that using air aeration to achieve the saturated dissolved oxygen value of water can meet the needs of the normal growth of plants, and excessively high dissolved oxygen did not promote the production.

Under this experimental condition, the higher the MNB concentration, the better the rice growth, the more developed the roots, the stronger the root activity, the more the nitrogen uptake and utilization, and the higher the grain yield. The performance of the HM treatment was better than the MM treatment and better than the LM treatment compared to CK both for the early rice and the late rice. In addition, we did not find the phenomenon described by Zhu et al. [33], who reported that excessively high oxygen concentration can inhibit the nitrogen absorption. This may be due to the fact that the MNB generator used in this study was an air-source isolated MNB pump, which is limited by power at a maximum air admission speed 1000 mL min$^{-1}$, producing a limited concentration of MNBs. Using an oxygen-source MNB generator increases the MNB concentration. Which MNB concentration of early rice and late rice is optimal and whether the high MNB concentration has a toxic effect on rice is worth further research. Simultaneously, this study also found that the average diameter of rice decreased with increasing MNB concentration both in early and late rice. This is similar to the report of Zhao et al. [28]. This may be the result of

adaptive regulation of the rice rhizosphere, which reduces the absorption of high MNBs by reducing the root diameter. Whether this is a general rule is worth further study.

This study was carried out under the ideal conditions and no external interference, such as rain, in order to discover the optimal concentration of MNBs required by double cropping rice accurately. If the same results are obtained by extending it to the rice field still needs further study.

As a new technology, MNB irrigation technology is in the exploratory stage. The cost of the MNB generator is still expensive, and the effect of MNB irrigation technology on the crops can easily be affected by the environment, which has caused difficulties in applying this technology on a large scale. However, with the progress of science and technology, it is believed that the cost of the MNB generator will become cheaper and the MNB irrigation technology will be less restricted by the external environment. This way, the optimal amount of MNBs which the crops need can be provided, and this technology will be used by more and more farmers to produce more grain yield with less water and fertilizers.

## 5. Conclusions

In summary, our results indicated that three concentrations of MNBs obviously enhanced the dissolved oxygen content of the solution of the paddy rice soil, promoted the growth and development of various organs, improved the nitrogen absorption and utilization, and increased the grain yield of double cropping rice. The MNB concentration of the HM treatment (air admission speed 900 mL min$^{-1}$) was the optimal concentration both for the early and late rice. The effect of MNBs on the late rice was better than that of early rice.

**Author Contributions:** Y.Q. and C.P. designed the experiments; X.G., C.S. and C.Q. performed the experiments; X.C. and J.C. analyzed the data; Y.Q. wrote the manuscript. All authors have read and agreed to the published version of the manuscript.

**Funding:** This work was supported by the National Key Research and Development Project of China (2016YFD0801101, 2018YFD0800503), the National Natural Science Foundation of China (32060431), and the Basic Research and Talent Training Special Project of Jiangxi Provincial Academy of Agricultural Sciences (JXSNKYJCRC202214).

**Institutional Review Board Statement:** Not applicable.

**Informed Consent Statement:** Not applicable.

**Data Availability Statement:** Not applicable.

**Acknowledgments:** The authors thank Xihuan Liang, Jiang Xie, and Guoqiang Deng for their participation in the study.

**Conflicts of Interest:** The authors declare no conflict of interest.

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
