# Peer review of "Effects of Different Concentrations of Micro-Nano Bubbles on Grain Yield and Nitrogen Absorption and Utilization of Double Cropping Rice in South China"

_agronomy, doi:10.3390/agronomy12092196_

Round 1
Reviewer 1 Report
1- In the first of the Abstract start with the issue and rationale.
2- The introduction needs improvement, the hypothesis, rationale and objectives are unclear.
3- Figure 1, What is the meaning of LM,MM,HM.... Please explain in the caption. The same in Tables
4- Merge Figure 2 with Figure 3 and use four panels as A,B,C, and D
5- In the merged Figure, explain in the caption what are the bars and error bars refer to
6- Merg Figure 4 with figure 5
7- In general, Results section looks weak, try to improve it with additional analysis (i.e. Correlation matrix, PCA...)
Author Response
Response to Reviewer 1 Comments
Point 1: In the first of the Abstract start with the issue and rationale.
Response 1: Thanks for your good suggestion.we had added the issue and rationale at the begining of the Abstract.“Micro-Nano Bubbles (MNBs) irrigation can effectively improve the hypoxia stress caused by conventional irrigation and show great potential in plant development, yield improving and water and fertilizer saving, and has been recognized as a new and high-efficiency technology in crop planting. However, the former researches of MNBs concentration had no clear segmentation and other MNBs con-centration can achieve better or worse effect? it remains to be further explored.”
Point 2- The introduction needs improvement, the hypothesis, rationale and objectives are unclear.
Response 2: Thanks for your good suggestion. Our main objectives of this paper were (1) to investigate the effect of the different MNBs concentrations on the agronomical, physiological and nutritional traits of the double cropping rice and (2) to investigate which is the optimal MNBs concentration for the double cropping rice. And this work had not be studied yet. the effects of different MNBs concentrations have different effects on the growth of the rice in our practices.it is necessary to know which is the optimal MNBs concentration in order to make us use the MNBs better. the probably reason for different MNBs concentration had different effects on the rice maybe lies in different MNBs concentrations have different oxygen contents that change the enviropment in which the rice grows but it is not clear and needs to study. we added some content in the introduction parts in order to make the reason why we studied this work to be more clear.
Point 3- Figure 1, What is the meaning of LM,MM,HM.... Please explain in the caption. The same in Tables
Response 3: Thanks for your good suggestion,we had explain the meaning of LM,MM,HM in the Figure 1 caption as suggeted.
Point 4- Merge Figure 2 with Figure 3 and use four panels as A,B,C, and D
Response 4: Thanks for your good suggestion, we had merged Figure 2 with Figure 3 into Figure 2 and use four panels as A,B,C, and D.
Point 5- In the merged Figure, explain in the caption what are the bars and error bars refer to
Response 5: Thanks for your good suggestion, we had explained what are the bars and error bars refer to in the Figure 2 caption.
Point 6- Merg Figure 4 with figure 5
Response 6: Thanks for your good suggestion, we had merged Figure 4 with Figure 5 into Figure3.
Point 7- In general, Results section looks weak, try to improve it with additional analysis (i.e. Correlation matrix, PCA...)
Response 7: Thanks for your good suggestion,we added section 3.7 and used the Correlation matrix to analy the relation between the main characteristics and the grain yield .

Reviewer 2 Report
The findings of this investigation are interesting, and although the experience has been carried out in pots, that is, in the laboratory, which is normal for an investigation, it is worth considering if the same results are obtained by extending it to the field. The question can also be posed as follows: to what extent is the effect of the MNB on the crop diminished when it is taken from the laboratory to the crop plot?
It is also worth asking if farmers are prepared to carry out this cultivation technique. All this, considering that the cultivated varieties are the same.
Although there is an issue that deserves an explanation. In the text, the authors indicate that the experience is carried out indoors, that is, in a greenhouse. However, in line 123, it is stated that rain must be avoided during the rice growing season. Does this situation endanger this cultivation technique? Because it cannot be moved or carried out outdoors.
I would recommend that, in the text, when the results of the repetitions are analysed or presented, they are separated by "semicolons" for example in the lines: 108; 176; 178, 193; 194; 209-212; 226-227; 236; 247-249.
I would also recommend that they highlight a little more what they contribute to scientific knowledge with their research, since little research has been carried out with this cultivation technique, and specifically in a crop, such as rice, which feeds half of the world population, as the authors point out.
Author Response
Response to Reviewer 2 Comments
Point 1: The findings of this investigation are interesting, and although the experience has been carried out in pots, that is, in the laboratory, which is normal for an investigation, it is worth considering if the same results are obtained by extending it to the field. The question can also be posed as follows: to what extent is the effect of the MNB on the crop diminished when it is taken from the laboratory to the crop plot?
Response 1: Thanks for your good comments. In order to know which is the optimal concentration of the MNBs for the double cropping rice,we conducted this experiment. the main reason we carried out this experiment in pots under the greenhouse instead of in the rice paddy is that there are many uncontrollable factors in field production, such as rain, etc. these uncontrollable factors first increased the difficulties of the study, and second has a great influence on the experimental results.
this is our first step to know the effect of the MNBs concentrations on the double cropping rice under an ideal conditions, and the result was the high MNBs concentration got the highest yield. the next step we will conduct the experiment in the rice paddy as you suggested to test whether or not the results are the same in the field conditions. and what extent of the different MNBs concentration on the double cropping rice diminished when it is taken from the greenhouse to the rice paddy. and it is a very important step we need to do in the further studies.
Point 2: It is also worth asking if farmers are prepared to carry out this cultivation technique. All this, considering that the cultivated varieties are the same.
Response 2: Thanks for your good comments. MNBs irrigation is a new technology and still in the exploratory stage. the cost of MNBs generator is still expensive, and the effect of MNBs irrigation technology on the crops can easily affected by the environment, which have caused difficulties in applying and broadcasting this technology in a large scale. But with the progress of science and technology. It is believed that the cost of the MNBs generator will be very cheap and the MNBs irrigation technology will be less restricted by the external environment, and can provide the optimal amounts of MNBs which the crops needed, and this technology will produce more grain yield with less water and fertilizers. and the farmers will pleased to use and broadcast this technology.
Point 3: Although there is an issue that deserves an explanation. In the text, the authors indicate that the experience is carried out indoors, that is, in a greenhouse. However, in line 123, it is stated that rain must be avoided during the rice growing season. Does this situation endanger this cultivation technique? Because it cannot be moved or carried out outdoors.
Response 3: Thanks for your careful reading. In order to know the effects of the different MNBs concentrations on the double cropping rice accurately ,we conducted this experiment under an ideal conditons without any external factors such as rain.etc. the rain will not endanger the MNB irrigation technique but will change the concentration of the MNBs in the soil, and may affect the result. so the rain must be avoied during the rice growing season.
Point 4: I would recommend that, in the text, when the results of the repetitions are analysed or presented, they are separated by "semicolons" for example in the lines: 108; 176; 178, 193; 194; 209-212; 226-227; 236; 247-249.
Response 4: Thanks for your careful work and give us very good suggestion. We had changed the “comma “into “semicolons”as you suggested.
Point 5: I would also recommend that they highlight a little more what they contribute to scientific knowledge with their research, since little research has been carried out with this cultivation technique, and specifically in a crop, such as rice, which feeds half of the world population, as the authors point out.
Response 5: Thanks for your great advice. We mentioned that” Rice (Oryza sativa L.) is the world’s most important food crop and a primary source of food for more than half the world’s population “ and added the”China accounts for about 20% of the world's rice acreage.” and”Only a few reports on the effect of the paddy rice” ,”However, these experiments of MNBs concentration had no clear segmentation and other MNBs concentration can achieve better or worse effect on the growth of paddy rice, especially in double cropping rice? While double-rice cropping is a typical farming system in southern China, accounting for 1/3 rice planting area in China in 2021. The double-rice cropping system typical-ly consists of early rice and late rice.and which is the optimal MNBs concentration for the double cropping rice has not known yet. Will high concentration MNBs led to negative effects on the rice growth? it remains to be fur-ther explored.” in the introduction.

Round 2
Reviewer 1 Report
Accept